# HPV-Associated Benign Squamous Cell Papillomas in the Upper Aero-Digestive Tract and Their Malignant Potential

**DOI:** 10.3390/v13081624

**Published:** 2021-08-17

**Authors:** Stina Syrjänen, Kari Syrjänen

**Affiliations:** 1Department of Oral Pathology and Oral Radiology, Institute of Dentistry, Faculty of Medicine, University of Turku, 20520 Turku, Finland; 2Department of Pathology, Turku University Hospital, 20521 Turku, Finland; 3Department of Clinical Research, Biohit Oyj, 00880 Helsinki, Finland; kasyrja@sci.fi

**Keywords:** papilloma, oral cavity, paranasal sinus, sinonasal, oropharynx, nasopharynx, larynx, esophagus, human papillomavirus, malignant transformation, risk factor, aneuploidy, transmission

## Abstract

Squamous cell papilloma (SCP) in the upper aero-digestive tract is a rare disease entity with bimodal age presentation both at childhood and in adults. It originates from stratified squamous and/or respiratory epithelium. Traditionally, SCPs have been linked to chemical or mechanical irritation but, since the 1980s, they have also been associated with human papillomavirus (HPV) infection. Approximately 30% of the head and neck SCPs are associated with HPV infection, with this association being highest for laryngeal papillomas (76–94%), followed by oral (27–48%), sinonasal (25–40%), and oropharyngeal papillomas (6–7%). There is, however, a wide variation in HPV prevalence, the highest being in esophageal SCPs (11–57%). HPV6 and HPV11 are the two main HPV genotypes present, but these are also high-risk HPVs as they are infrequently detected. Some 20% of the oral and oropharyngeal papillomas also contain cutaneous HPV genotypes. Despite their benign morphology, some SCPs tend to recur and even undergo malignant transformation. The highest malignant potential is associated with sinonasal inverted papillomas (7–11%). This review discusses the evidence regarding HPV etiology of benign SCPs in the upper aero-digestive tract and their HPV-related malignant transformation. In addition, studies on HPV exposure at an early age are discussed, as are the animal models shedding light on HPV transmission, viral latency, and its reactivation.

## 1. Introduction

The upper aero-digestive tract is composed of the nasal cavity, paranasal sinuses, nasopharynx, oral cavity, oropharynx, hypopharynx, larynx, trachea, and esophagus. Squamous cell papillomas (SCPs) are histologically benign growths encountered practically at all body sites where squamous epithelium exists (e.g., skin, eye conjunctiva, paranasal sinuses, pharynx, oral cavity, larynx, esophagus, bronchus, genital tract, urinary tract) [1]. SCPs are composed of either exophytic and/or papillary fronds with fibrovascular cores or endophytic epithelial invaginations lined by stratified squamous or respiratory epithelium, depending on their anatomic site of origin.

In the aero-digestive tract, the most common sites of SCPs are the oral cavity and the larynx, while sinonasal and esophageal areas are less commonly involved. Only few SCPs have been described in the naso- and oropharynx (especially base of the tongue and tonsils) [1,2,3,4,5,6,7,8]. This might relate, however, to the ease of detection of these lesions, as oral papillomas are visible to the naked eye and even tiny laryngeal lesions can cause clinical symptoms (hoarseness) earlier than SCPs at other anatomic sites.

The etiology of SCP is not universally confirmed but mechanical and chemical irritation and/or infection with human papillomaviruses (HPV) have been prime culprits [1]. The most widely studied entities are sinonasal papillomas and recurrent respiratory papilloma (RRP) of the larynx. This is because of their more aggressive behavior and potential for malignant transformation. Aggressive clinical course (i.e., recurrence) of the disease is found in 20% of the juvenile-onset-RRP (JO-RRP) patients, while malignant transformation is most prevalent among patients with inverted sinonasal papillomas (7–11%), followed by adult-onset RRP (AO-RRP) (3–6%).

The two most frequent HPV genotypes associated with SCPs are HPV6 and HPV11, but high-risk HPV16 and 18, and other mucosal alpha-HPVs, have been detected at a lower frequency [1,2,9,10,11]. Importantly, recent evidence also indicates the presence of cutaneous HPVs of the beta-and/or gamma genera in oral, oro-, and hypopharyneal papillomas, as well as anecdotally in laryngeal papillomas [12,13]. Based on the animal studies, papillomavirus (PV)-associated SCPs are highly contagious lesions that produce viral particles, which are shed via the saliva and other secretions, allowing for the autoinoculation or horizontal transmission of the virus [14]. In autoinoculation, viral particles can infect adjacent traumatized mucosa, resulting in multiple lesions and/or the spreading of infection to other body sites (e.g., from sinonasal tract and/or oral cavity to naso- and oropharynx, larynx or vice versa). Maternal HPV transmission is closely associated only with JO-RRP but is not widely discussed in the context of pediatric SCPs located elsewhere in the aero-digestive tract [15,16].

This review summarizes the key features of SCPs in the upper aero-digestive tract (Figure 1), including the trachea and esophagus. We discuss the causative role of HPV infection and the site-specific differences in its aggressive behaviors and malignant potential for each anatomic location separately, highlighting the gaps in our knowledge. In addition, a brief account is examined in relation to animal models, HPV latency, and the successful eradication of RRP in Australia by the nationwide HPV vaccination program.

## 2. Sinonasal Papilloma (SNP)

Sinonasal papilloma (SNPs), identified in the 1850s has frequently been called the Schneiderian papilloma. This is because these lesions develop from the Schneiderian membrane, i.e., an ectodermally derived respiratory mucosa covering the nasal cavity and paranasal sinuses, in contrast to the other mucosa of the upper respiratory tract, which is derived from an embryological endoderm. The latest (2017) WHO classification omitted the eponymous “Schneiderian” designation in favor of the more simple sinonasal papilloma (SNP) [1]. SNPs comprise 0.5–4% of all nasal tumors, with an estimated annual incidence of 0.74–2.3 per 100,000 individuals [1,17]. In children, SNP is exceptionally rare, and only a few cases have been reported thus far [18,19,20,21].

The symptoms of SNP include unilateral nasal obstruction, epistaxis, bloody nasal discharge, headache, facial pain, anosmia, dysosmia, and epiphora. SNPs are characterized by three typical features: (1) a locally destructive growth pattern, (2) development of frequent recurrences, and (3) the potential for malignant progression [1,22,23,24]. Based on the classical studies by Buchwald in Denmark, the average incidence of SNP associated with carcinoma was calculated as 0.38 cases per million inhabitants per year (i.e., 36 cases in 19 years among 5 million inhabitants). As the annual incidence of squamous cell carcinoma (SCC) of the sinonasal cavities is 2.5 cases per million, 15% of these sinonasal SCCs are, according to this study, associated with SNPs [17,23,24].

### 2.1. Subtypes of SNPs

There are three histological subtypes of SNP: (1) inverted SNP, (2) exophytic SNP (previously known as fungiform papilloma), and (3) oncocytic SNP (also called cylindrical cell or columnar cell papilloma) [1,23,25].

Sinonasal inverted papilloma (SNIP) is the most commonly diagnosed subtype, accounting for almost 65–75% of all SNPs [1,19,25,26]. The tumor is most frequent in the 5th–6th decades and is three times more common in men than in women. The term “inverted papilloma” describes the epithelial growth pattern inward into the underlying supportive tissue of the sinonasal mucosa, which is characteristic of this particular tumor, as illustrated in Figure 2a [1]. SNIP differs markedly from the two other subtypes in its invasiveness, high recurrence rate, and association with sinonasal SCC. In 7–11% of cases, SNIP undergoes malignant transformation, which is synchronous in approximately 70% of cases [1,24,27]. These SCCs arising from SNIP have an aggressive loco-regional tendency, whereas de novo SCCs present with a higher propensity for aggressive distant metastases [1,24,27]. Age, tumor stage, and positive surgical margin are the key determinants of poor prognosis [28,29].

Sinonasal exophytic papillomas (SNEP) account for 20–25% of all SNPs [1,22,25], and they occur mostly in the 3rd to 5th decades. The majority of SNEPs are localized on the septal mucosa, growing as single unilateral lesions. Their light microscopic appearance is identical to SCPs at other mucosal sites, e.g., oral cavity and larynx, characterized by branching fronds of mucosa, covered by stratified squamous epithelium and supported by connective tissue scaffold (Figure 2b). Depending on the localization, the squamous epithelium may range from immature (basal type) to fully keratinized epithelium, particularly in the nasal vestibulum [1].

Sinonasal oncocytic papilloma (SNOP) is the most uncommon type, representing only 3–6% of all SNPs [1,22,25]. Its location is typically the lateral nasal wall or paranasal sinuses, with maxillary sinus being the most common (60%) site. Both genders are affected equally. SNOPs are characterized by both inverted and exophytic growth pattern but differ from the other papilloma subtypes by their pseudostratified columnar epithelial lining, with cells containing abundant, granular eosinophilic cytoplasm and small, hyperchromatic nuclei. Another characteristic feature is the frequent presence of numerous intraepithelial micro-cysts. A recent study from Finland comprising 20 patients with SNOPs diagnosed during 1994–2016, showed that 39% of these lesions recurred, the recurrence being more frequent in tumors located in the sinuses than in those of the nasal cavity (45% vs. 29%). Interestingly, recurrent lesions were more common among smokers than nonsmokers (75% vs. 31%), but none of the lesions presented with any degree of dysplasia [30].

### 2.2. Etiology of Sinonasal Papillomas

The etiology of SNPs is not yet universally accepted. Several risk factors have been identified, including cigarette smoking, alcohol consumption, allergic rhinitis, essential hypertension, anticoagulant medication, as well as exposure to environmental and occupational noxious agents such as organic solvents, welding fumes, and nickel compounds [26,31,32]. Sham et al. (2010) could confirm only outdoor and industrial occupations as the risk factors of SNIP, whereas tobacco smoking, drinking alcohol, history of allergic rhinitis, sinusitis, and nasal polyps were not considered as risk factors [32]. Similarly, Pähler vor der Holte et al. (2020) reported that tobacco smoking was not a risk factor for any of the three SNP subtypes or their recurrence in a German cohort of 154 SNPs [2].

#### HPV in Sinonasal Papilloma

Since pioneering original observations in the early 1980s, implicating HPV as the potential etiological agent of SNPs, the discussion is still ongoing regarding the etiological role of HPV in SNPs and their malignant transformation. This discussion has been updated in recently published systematic reviews and meta-analyses.

In 2013, we published a systematic review and meta-analysis of the literature reporting on HPV detection in SNPs up until April 2012. Seventy-six studies were eligible, covering 1956 SNPs from different geographic regions. Altogether, 38.8% (*n* = 760) cases were HPV-positive, with HPV prevalence being highest (65.3%) in SNEPs, followed by SNIP (37.8%) and SNOP (22.5%) [33]. The likelihood of detecting HPV in exophytic papilloma was 34.6-fold higher than in normal sinonasal mucosa (OR = 34.64; (95% CI, 18.19–68.35). Similarly, the likelihood of detecting HPV in SNIP was nearly 12-fold higher than in normal mucosa (OR = 11.83; 95% CI, 6.97–21.59), whereas HPV had no significant association with SNOP (OR = 3.07: 95% CI, 0.68 to 10.59). Neither HPV testing method nor geographic origin of the study was an independent study-level covariate in formal meta-regression [33].

Gupta et al. (2020) summarized the results from 21 additional studies published since the appearance of our meta-analysis [33]. Of the SNIPs, 23.3% (330/1416) were HPV-positive but the positivity varied from 0% to 62% in different studies. Some studies also stratified the analyses of SNIP by histology, separating non-malignant (NM-SNIPs) from those with dysplasia (D-SNIPs), as well as those with concurrent malignancy (M-SNIPs). In total, 20.9% (228/1092) NM-SNIPs, 52.6% (*n* = 40/76) D-SNIPs, and 23.6% (29/123) M-SNIPs tested HPV-positive, thus disclosing the highest HPV prevalence (>50%) in SNIP with concomitant dysplasia [34].

Recently, Pähler Vor der Holte et al. (2020) reported that 31 of 80 SNIPs were HPV- positive (38.8%). The most commonly detected HPV genotypes were HPV6 (21/80, 26.3%) and HPV16 (18/80, 22.5%), followed by HPV11 (in 10/80, 12.5%), HPV58 (4/80, 5%), HPV42, and HPV83 (one case each, 1.3%). Most patients displayed an infection with just one HPV type. In total, 16.3% of the SNIPs had a co-infection with two HPV types, but one patient had co-infection with four different HPV types. In that series, only four SNEPs were available, all being HPV-positive: HPV6, HPV11, and HPV 91. Of the two SNOPs, the larger one, spanning three paranasal sinuses, presented co-infections with HPV6, HPV16, and HPV42 [2].

The role of cutaneous β- and γ-HPVs in SNPs is practically unexplored, although evidence is available that these viruses are present as asymptomatic infections both in the nasal cavity and in oral mucosa [12,13]. Due to the importance of this topic, we will shortly review the study on cutaneous HPVs by Forslund et al. [35]. In their series of 311 oral and 304 nasal samples collected from 312 volunteer Danish healthcare staff, HPV DNA was found in 6% and 50% of the oral and nasal samples, respectively. No gender difference in HPV prevalence was found. Altogether, 75 diverse HPV genotypes or putative HPV genotypes were identified. HPV genotypes within the α-, β-, and γ-HPV genera were detected in 3%, 31%, and 23% of the nasal samples, respectively. The detected β -HPVs were within species β-1, β-2, β-3, and β-5 in 17%, 9%, 7%, and 1% of the nasal samples, respectively [35]. An interesting putative subtype of HPV76, originally isolated from a feline oral squamous cell carcinoma, was detected in seven nasal samples. Thus, a large spectrum of HPV genotypes from β-HPVs and γ-HPV genera seems to demonstrate tropism to the nasal mucosa. As many cutaneous HPV genotypes identified in normal nasal mucosa have been associated with skin carcinomas, there is an urgent need to study the role of β -and γ-HPV genera in SNPs and SNP-derived SCCs.

### 2.3. Malignant Transformation

The estimated overall rate of malignant transformation of SNPs is approximately 9% (95% CI 7–11%), based on the systematic review by Re et al. (2017), who included 29 studies with a total of 3177 patients [36]. The risk factors of malignant transformation are nearly the same as those listed for SNPs: outdoor and industrial occupations, exposure to organic solvents in particular, e.g., diethyl nitrosamine, as well as cigarette smoking and HPV. Katori et al. (2006) analyzed the histological features of SNPs that could predict their malignant transformation [37]. Malignancy was related to the presence of bone invasion (*p* = 0.039), the absence of an inflammatory polyp (*p* = 0.033), an increase in the ratio of neoplastic epithelium to stroma (*p* = 0.037), an increase in hyperkeratosis (*p* = 0.030), and a decrease in the number of eosinophils (*p* = 0.037) [37]. We have previously reported that DNA aneuploidy might predict the malignant transformation of SNPs [38].

#### 2.3.1. Malignant Transformation and HPV

Lawson et al. (2008) reviewed the SNP literature published between January 1987 and December 2006 to address the following three important questions: (1) Why do HPV detection rates vary so greatly? (2) What is the relationship between LR- and HR-HPV types and HPV detection rates? (3) Is there a relationship between certain HPV genotypes and lesion recurrence or their malignant progression? [39]. The first important conclusion was that the detection rates were similar across HPV detection methods: 26.8% (95% CI 16.4–37.2%) by in situ hybridization (ISH), 25.2% (95% CI 14.7–35.6%) by consensus PCR, and 23.6% (95% CI 12.2–35.0%) by type-specific PCR. HPV 6/11 was more prevalent in SNPs than HPV 16/18. The overall unadjusted ratio of LR-HPV to HR-HPV types was 2.8:1 [39].

HPV detection rates significantly increased (*p* < 0.02) in SNIP with high-grade dysplasia (WP 55.8%, 95% CI 30.5–81.0%) and SCC (WP 55.1%, 95% CI 37.0–73.2%) compared to SNIP with no dysplasia or mild dysplasia (WP 22.3%, 95% CI 15.9–28.6%). Furthermore, the preponderance of LR-HPV in benign SNIP (ratio LR/HR = 4.8:1) shifted in dysplastic and malignant SNPs. The LR/HR ratio was 1.1:1 for SNIPs with high-grade dysplasia, but reverted in favor of HR-HPV (1:2.4) in SNIPs with malignant transformation [39]. Recurrences developed in 44/236 patients and HPV was significantly associated with the likelihood of developing a recurrence (weighted OR = 10.2, 95% CI 3.2–32.8) [39].

Ding et al. (2020) authored a systematic review and meta-analysis on malignant transformation in SNIP stratified by HPV genotypes [40]. Altogether, 26 studies with 900 SNIP patients were eligible for this the meta-analysis. The pooled ORs indicated that HPV6 (OR: 2.02; 95% CI: 0.47–8.61; *p* = 0.343), HPV11 (OR: 0.86; 95% CI: 0.26–2.89; *p* = 0.806), and HPV 6/11 (OR: 1.44; 95% CI: 0.59–3.53; *p* = 0.426) infections were not significantly associated with the risk of malignant SNIP. In contrast, the risk of malignant SNIP was significantly increased in patients infected with HPV16 (OR: 8.51; 95% CI: 3.36–21.59; *p* < 0.001), HPV11/16 (OR: 7.95; 95% CI: 1.13–56.01; *p* = 0.038), HPV18 (OR: 23.26; 95% CI: 5.27–102.73; *p* < 0.001), and HPV16/18 (OR: 24.34; 95% CI: 5.74–103.18; *p* < 0.001) [40].

Another meta-analysis by Stepp and coworkers (2021) focused on the role of HPV in the malignant transformation of SNIP [41], which was based on 19 high-quality case–control and cohort studies. The presence of HPV was associated with statistically significantly higher OR for malignant transformation of SNIP (unweighted, pooled OR = 2.38, 95% CI 1.47–3.83). HR-HPV types had a higher OR for SNIP-derived and SCC-derived compared to LR-HPV types (OR = 3.42, 95% CI 1.42–8.25; I2 = 39.1%) [41]. When the publication date was analyzed in 10-year blocks (1990–1999; 2000–2009; 2010–present), ORs did not reach statistical significance (*p* = 0.905), indicating that there was not a similar increase in HPV-associated SCC in sinonasal tract such as that found in HPV-associated oropharyngeal carcinomas. Importantly, the positive association of HPV with malignant SNIP transformation was confirmed in all three geographic regions of the study: North America, Europe, and Asia [41].

#### 2.3.2. Malignant Transformation and Genetic Profile

Molecular mechanisms underlying SNP, or their malignant progression, are still incompletely understood, and the number of cases studied is relatively small. This notwithstanding, the results are consistent and merit review in this paper [42,43,44,45]. The main results indicate that *EGFR* mutations and HR-HPV infection represent essential, alternative oncogenic mechanisms in SNIP and SNIP-associated SCC. Furthermore, alterations in *TP53* and *CDKN2A* are likely to be early markers for malignant progression of SNIP. Thus, there is a need for molecular sub-classification of SNP and sinonasal carcinomas because it may contribute to prognostic predictions and the design of personalized molecular-targeted precision medicine.

Sahnane et al. (2019) genotyped 10 genes involved in EGFR signaling in 25 SNIP, 5 SNOP, and 35 SCC samples from 54 patients [42]. Additionally, methylation of *LINE-1* was determined, as its hypo-methylation is known to be significantly associated with squamous histology, tobacco smoking, and poor prognosis. All SNOP lesions were HR-HPV-negative but had *KRAS* mutations, while all SNIPs were also negative for HR-HPV but had no *KRAS* mutations. HR-HPV was observed in only 13% of SNIP-associated SCC and in 8% of de novo-SCC patients. *EGFR* mutations were more common in SNIP than in SNIP-associated SCC (72% vs. 30%) or de novo-SCCs (17%) [42]. At 5-year follow-up, SCC developed in only 30% (6/20) of the patients with *EGFR*-mutated ISPs compared to 76% (13/17) of the patients with *EGFR*-wild-type ISP (*p* = 0.0044). *LINE-1* hypomethylation significantly increased from papilloma to early stage SCC, and further to advanced stage SCC (*p* = 0.03), and it was associated with occupational exposure (*p* = 0.01) and worse prognosis (*p* = 0.09) [42].

Hongo et al. (2020) determined the prevalence, mutual relationships, and clinic-pathological significance of *EGFR* mutation, *EGFR* copy number gain (CNG), *KRAS* mutation, and HR-HPV infection in 146 SNIP-associated SCCs [43]. HR-HPV was detected in 11 cases (7.5%), whereas all 14 SNIP-SCCs were negative. *EGFR* mutations were present in 21 (14.7%) of 143 SNIP-SCCs, including 13/14 (92.9%) SNIP-SCCs and 8/129 (6.2%) non–SNIP-SCCs (*p* < 0.0001). The majority of *EGFR* mutations were exon 20 insertions, with the remainder composed of deletions and single-nucleotide substitutions in exons 19 and 20. *KRAS* mutation was not detected in any of the 142 SNIP-SCCs. *EGFR* CNG was detected in 41 (28.1%) of 146 SNIP-SCCs; all of them were HPV-negative [43]. Collectively, *EGFR* mutation, *EGFR* CNG, and HR-HPV were essentially mutually exclusive, and each subgroup had distinct clinic-pathological features. The HPV-negative/*EGFR*-mutant group, the HPV-negative/*EGFR* CNG-positive group, and the triple-negative group had significantly worse prognoses than the HPV-positive group (*p* = 0.0265, *p* = 0.0264, and *p* = 0.0394, respectively). Thus, Hongo et al. (2020) confirmed the results by Sahnane et al. (2019), in that *EGFR* mutation may play a pathogenically important role in a subset of SNIP-SCCs [42,43].

In yet another study, Brown et al. (2021) characterized mutations and copy number alterations in 29 SNP-associated SCCs utilizing targeted next-generation DNA sequencing (DNAseq) of frequently altered pan-cancer genes [44]. They also evaluated molecular alterations within 11 matched SNP and SNP-SCC pairs and compared the molecular landscape of the SNP-SCC to the data available from large cohorts of the Cancer Genome Atlas (TCGA) for aero-digestive tract SCC, which comprises head, neck, and lung carcinomas [44].

According to these data, the vast majority of SNP-SCC (72.4%) showed evidence of at least one *CDKN2A* alteration, and all except one (96.6%) harbored at least one *TP53* or *CDKN2A* alteration. Importantly, none of these alterations were detected in the subset of matched SNP, indicating that *TP53* and *CDKN2A* are likely to be early molecular events in the malignant progression to SNP-SCC. *EGFR* and *KRAS* mutations were significantly enriched in SNP-associated SCC relative to other aero-digestive tract SCC (*p* < 0.001); *CDKN2A* mutations, *TERT* copy number gains, and LR--HPV infection also occur more frequently in these tumors (*p* < 0.05) [44]. These results confirm that SNP-associated SCCs are molecularly distinct from SCCs of the aero-digestive tract.

In a previous study, the same research group also showed that *EGFR* mutations (detected by Sanger sequencing) and HPV infections represent alternate oncogenic pathways [45]. That cohort included 58 cases of SNIP, 22 SNIP-associated SCC, and 14 SCC without SNIP. All SNIP and SNIP-associated SCCs demonstrated either an *EGFR* mutation or an HPV infection. Similarly, as reported by Sahnane et al. (2019) and Hongo et al. (2020), HPV and *EGFR* mutation was mutually exclusive in all cases of SNIP-associated SCC and all but one SNIP [42,43,44]. HPV genotypes detected in SNIP and SNIP-associated SCCs were predominantly of LR-type, in contrast with SCCs without SNIP, which were associated with HR-HPV genotypes. All paired SNIP and SNIP-SCC samples demonstrated concordant HPV status and *EGFR* genotypes. Malignant progression of SNIP was significantly associated with the presence of HPV infection and the absence of an *EGFR* mutation (log-rank = 9.620, *p* = 0.002) [45].

## 3. Nasopharyngeal Papillomas

At birth, nasopharynx is lined by typical respiratory epithelium, but this pseudostratified columnar ciliated epithelium is gradually replaced by stratified, non-ciliated epithelium and, with advancing age, by mature squamous epithelium. A transformation zone with distinct squamo-columnar junctions is typical of the nasopharynx; thus, one would expect to find this area to be a typical site for SCP. However, primary nasopharyngeal papilloma is an extreme rarity, as determined from the literature. Only 12 well defined “old” cases could be derived from the archives of The Armed Forces Institute of Pathology (AFIP) [4]. These lesions had a typical morphology of SNP, particularly the inverted type with squamous epithelium. All these cases were benign, with no tendency for recurrence or subsequent malignancy. HPV data were not available at that time.

The existent of nasopharyngeal papilloma is also ignored in the latest WHO book, 2017 [1]. Recently, Dona et al. (2017) analyzed the presence of HPV in respiratory papillomas, of which three originated from the nasopharynx [12]. Two of these nasopharyngeal papillomas tested positive for HPV6 and HPV11, and one for HPV93 (belongs to β-HPVs) [12]. The reasons for the reported rarity of papillomas arising in the nasopharynx remain obscure, but they might be explained by the fact that these lesions have been either overlooked or classified as SNPs (with posterior location) in these reports. Beyond doubt, further studies are needed to clarify the prevalence of nasopharyngeal SCPs and their association with HPV.

## 4. Oral Squamous Cell Papilloma (OP)

The oral cavity is unique in its anatomy and dual function, comprising the first portion of the digestive tract but also part of the upper airways (Figure 1). Depending on the site, oral epithelium varies from (i) completely keratinized with a superficial horny layer (surface of tongue, hard palate, and gums) to (ii) parakeratinized or (iii) non-keratinized squamous epithelium (e.g., buccal mucosa).

Oral squamous cell papilloma (OP) is still a controversial issue. The discussion is ongoing as to whether these lesions represent a reaction to injury rather than a true benign neoplasm [1]. In animal studies, the PV-etiology of these lesions is well established, as is the viral latency in oral mucosa [14,46], which will be discussed later in paragraph 10. Population-based studies on the prevalence and incidence of OP lack similarly in relation to recent systemic studies. The largest cohort study available is still one by Axell from 1976. He made a clinical examination of 20,000 Swedish citizens, of whom 0.1% had oral warty lesions (incidence of 0.5/100,000) [47].

Based on clinical examinations, Knapp et al. (1971) reported that OP was the second most common lesion among 181,338 consecutive oral examinations made in the US Army [4]. OPs occur at any age, but they are most frequently seen in patients in the 4th and 5th decades [3,47,48,49]. However, some 8% of the patients were younger than 10 (Figure 3). OPs are located most frequently on the palatal complex, dorsum, and lateral borders of the tongue, as well as in the lower lip. In this context, the palate complex needs a closer look because the soft palate and uvula are the sites of oropharynx. Abby et al. (1980) reported that OPs in the palatal complex accounted for 34.3%, but were more frequent in the soft palate than in the hard palate, with a ratio of 2:1. There is no gender differences among the OPs (male: female ratio of 1:1). Most of the lesions (87.5%) are found in Caucasians, and the majority seem to clear in one year, though persistence up to 10 years has been reported [3]. Recurrence and/or multiple lesions are rare (4.1%). A slight degree of epithelial cell atypia was present in 25% of the OPs, while extensive atypical features were rare (1.7%), based on the following criteria: abnormal mitoses, individual cell keratinization, and increased nuclear/cytoplasmic ratio [3]. Thus far, no reports on malignant transformation of OPs exists in the literature.

### 4.1. Multiple OPs and Genetic Syndromes

Some rare genetic syndromes are known to be associated with multiple OPs, including (1) focal dermal hypoplasia syndrome, (2) WHIM syndrome, (3) acrodermatitis enteropatica, (4) Cowden syndrome, (5) nevus unius lateralis, (6) Costello syndrome, and (7) Down syndrome [50]. WHIM syndrome is an immunodeficiency disease characterized by neutropenia, hypogammaglobulinemia and extensive HPV infection at different body sites. WHIM syndrome (WHIMS) is caused by heterozygous mutation in the CXCR4 gene on chromosome 2q22 [51]. Interestingly, multiple papillomas in the esophagus are associated with some of these syndromes, as are laryngeal papillomas with the Cowden syndrome. No systematic analyses of the associations of these papillomatous lesions with HPV are available, except for WHIM.

### 4.2. HPV and Oral Squamous Cell Papilloma

Frithiof and Wersall (1967), as well as Gysland et al. (1976), were the first to identify intra-nuclear viral particles closely resembling HPV in the epithelial cells of an OP and oral condyloma, respectively [52,53]. We reviewed the literature until 1998 and identified 481 OPs and 284 oral condylomas that were analyzed for HPV [54]. The overall detection rate for HPV DNA was 49.8% and 75% in OPs and oral condylomas, respectively, with HPV6 being the most prevalent genotype, followed by HPV11. Additionally, HPV16, HPV18, and HPV33 were detected, albeit rarely. However, the number of samples analyzed in individual studies was limited, and HPV DNA testing methods included mostly ISH, dot blot-, and Southern blot hybridization. At that time, only 44 samples were tested with PCR, which increased the detection rate up to 70.5% [54]. According to the current view, OP cannot be reliably distinguished from oral condyloma either clinically or by light microscopy [54].

Trzcinska et al. (2020) studied the presence of HPV in a series of 131 SCPs of the head and neck region, including 19 OPs [9]. Of the OPs, 21% (*n* = 4/19) were HPV-positive and HPV6 or HPV11 were the only genotypes detected. Danna et al. (2017) analyzed 83 SCPs derived from different sites of the respiratory tract (43 oropharyngeal, 31 oral, 6 laryngeal, and 3 nasopharyngeal) [12]. Twenty-four samples (28.9%) were positive for mucosal HPVs, of which three were oropharyngeal (6.9%), fifteen oral (48.3%), four laryngeal (66.7%), and two nasopharyngeal papillomas (66.7%). Among the 81 cases also tested for cutaneous HPV types, mucosal and/or cutaneous HPV types were detected in 43.2% of the cases, of which 13.5% harbored only cutaneous types, and 6.2% were positive for both cutaneous and mucosal HPVs. Of the LR-HPVs, only HPV6 was found in 29% (9/31) of the samples. HR-HPVs were detected in 16.1% (5/31), of which HPV16 was in 3 OPs and HPV18 and HPV35 were in 1 case each. In total, 22.6% of the OPs were positive for cutaneous types. The following β-HPVs were found: HPV5, HPV12, HPV120 present in one case each, and HPV23 present in two OPs. The following γ-HPVs were identified: HPV121, 123, 130, and 131, in one case each and also occasionally present as co-infections. The prevalence of α- and cutaneous HPVs were higher in oral than in oropharyngeal papillomas: 48.3% and 22.6% vs. 6.9% and 14.7%, respectively. Interestingly, in oropharyngeal lesions, cutaneous HPVs prevailed over mucosal HPVs [12].

We recently published data on 83 oral epithelial lesions, of which 33 (39.8%) were diagnosed as OP, 40 (48.2%) as papillary hyperplasia, 6 (7.2%) as verruca vulgaris, and 4 (4.8%) as others [13]. The overall positivity for HPV DNA was 21.7% (18/83). Only 6.4% (13/83) of the samples were positive for α-HPV, two (2.4%) for β-HPV, and four (4.8%) for γ-HPV. Among the α-, β, and γ-HPV, the following genotypes were found: HPV 6/11/16, 8/22, and 161/170, respectively. Of the oral epithelial lesions, OP was most commonly associated with HPV, and was present in 27.3% (9/33) of the samples. The mucosal HPV genotypes predominated in OPs as follows: 5 samples with either HPV6 or HPV11 and 2 samples with HR-HPV16. Of the cutaneous genotypes, HPV22 (β-HPV) and HPV161 (γ-HPV) were identified in one patient each [13].

As discussed in relation to SNIP, it is important to elucidate the role of cutaneous HPV genotypes in both benign and premalignant and malignant oral lesions. It is currently known that asymptomatic HPV infections with a wide spectrum of β- and γ-HPV genera are found in oral and oropharyngeal mucosa [39,55]. Bottalico et al. (2011) reported that rinse specimens from both oral and oropharyngeal areas of 35/52 (67%) HIV-positive individuals and 117/317 (37%) older male participants tested positive for HPV DNA. Type-specific HPV infections from the 52 HIV-positive individuals included 73 α-HPV, 33 β-HPV, and 11 γ-HPV infections, whereas the type-specific HPV infections in of the older HIV-negative males included 46 α-HPV, 108 β-HPV, and 14 γ-HPV infections [55].

Oral mucosa contains a wide spectrum of HPV genotypes, predominantly of the β-HPV and γ-HPV genera. Future studies are clearly warranted on the whole spectra of HPV genotypes present in oral mucosa to gain further insights into the biology of HPV and its association with oral neoplasia.

## 5. Oropharyngeal Papilloma

Unfortunately, the data are scant regarding oropharyngeal SCPs. Kim et al. (2019) authored a retrospective study on oropharyngeal papillomas diagnosed between 2013 and 2015 [6]. In their cohort, the discovery of oropharyngeal SCPs was incidental, the majority being found on workup for other unrelated symptoms, such as difficulty in swallowing, throat pain, or hoarseness [6]. However, most of them were asymptomatic, especially when small and slowly growing. A total of 26 cases were identified, 13 females and 13 males, with a median age of 58 years (range: 21–77). The most common locations were the base of tongue, uvula, tonsils, and the soft palate, thus confirming previous observations [3,6,12]. The papillomas removed in the office ranged from 0.2 to 1.5 cm in size (mean 0.9 cm), and 5/26 (19%) of the lesions recurred [6].

### HPV Prevalence

HPV presence in oropharyngeal papillomas has not been systematically studied, and the limited data discussed here are extracted from a few recent studies on SCPs with a focus on several sites of the respiratory tract. As already discussed, Dona et al. (2017) studied 83 papillomas, of which 43 were of oropharyngeal origin [12]. Only 3 of the 43 samples (6.9%) harbored mucosal HPV DNA, with 2 being positive for either HPV6 or HPV11 and 1 for the high-risk HPV51 and undetermined HPV74. Cutaneous HPVs were detected in 6 samples (14.7%), of which 4 were positive for β-HPVs (2 cases HPV12, 1 HPV 23, and 1 HPV 24) and 2 for γ-HPVs (HPV131 or HPV 156 [12]. In a retrospective series by Kim et al. (2019), only 16/26 samples were tested for HPV [6], with 75% being HPV-negative. Two tested positive for LR-HPVs, one for both LR- and HR-HPVs, and one with an unidentified HPV genotype. A recent study by Trzcinska et al. (2020) on HPV in head and neck SCPs identified 63 oropharyngeal papillomas, of which only 4 (6%) were HPV-positive; 2 with HPV6, 1 with HPV11, and 1 sample with an unidentified genotype [9].

## 6. Laryngeal Squamous Cell Papillomas (RRP)

Laryngeal papilloma (LP) is a disease with a long history, described as “warts in the throat” in the seventeenth century, and it was first recognized as an entity separate from other laryngeal lesions by MacKenzie (1871) [56]. Originally, four distinct types of LPs were distinguished on the basis of their onset, clinical appearance, and natural history: (a) juvenile solitary; (b) juvenile multiple; (c) adult solitary; and (d) adult multiple [57]. More recently, the nomenclature has been revised by the term “recurrent respiratory papillomatosis” (RRP), which more accurately describes the widespread extent of the disease and its tendency for repeated recurrence, and also encompasses both juvenile (JO-RRP)- and adult (AO-RRP)-onset forms of the disease [1,8]. In this discourse, we adopt these abbreviations, as well as LP, to stand for laryngeal papilloma when appropriate.

The estimated incidence of RRP varies from 0.6 to 4.3/100,000 among children and 1.8 to 2.3/100,000 in adults. [1]. Notably, the incidence of JO-RRP seems to be nearly six times higher in the USA (4.3/100,000) than in Europe (0.6–0.8/100,000) [8,26,58]. According to the bimodal incidence pattern of RRP, the first age peak is seen in children aged < 5 years (JO-RRP) and the second one in adults aged 20–40 years (AO-RRP). In children, there is no gender predominance, while in adults, the male–female ratio is 3:2 [1].

Clinically, LP is a multi-nodular, cauliflower-like exophytic tumor composed of stratified squamous epithelium with vascular connective tissue core, and it is similar to the histology of SCPs at other mucosal sites. The course of these benign lesions is highly variable; some patients have spontaneous remission, whereas, in others, morbidity is high because these lesions are difficult to eradicate due to their high tendency for recurrence and spread throughout the aero-digestive tract. In adults, RRP lesions are more often solitary and less severe, and upper airway obstruction occurs less frequently than in children. Malignant transformation into laryngeal (tracheal, bronchial) SCCs does occur, but rarely in a proportion of disease prevalence [59].

### 6.1. HPV6 and HPV11 Are the Major Causative Agents of RRP

Ulmann (1923) suspected PV etiology of LPs long before HPV particles could be demonstrated by electron microscopy [60]. He also successfully induced warts in dogs (and in his own arm) by transmitting non-filtered extract of a human LP. Hajek (1956) was the first to suggest the maternal association with LPs in children [61]. In 1982, Mounts et al. were able to identify the HPV6 genome in LPs of both adults and children [62]. In 1987, Byrne et al. described the presence of HPV11 in a patient with chronic laryngotracheobronchial papillomatosis and metastatic SCC of the lung [63]. In our HPV textbook [54], we summarized the data on HPV detection in LPs, covering the studies between 1980 and April 1998. At that time, 53 reports were identified providing the data from 688 LPs, including both AO-RRP and JO-RRP. HPV testing included methods with highly variable sensitivity, e.g., filter methods (dot blot and Southern blot hybridization), ISH, and subsequently PCR. In total, 76.2% (524/688) of LPs were HPV DNA-positive, almost exclusively for HPV6 and HPV11. Other HPV types such as HPV16, 18, 31, and 33 were reported in anecdotal cases [54].

Since 1998, an increasing number of studies on HPV presence in RRP have been published, and the proportion of HPV positivity in RRPs has increased close to 90–95%, with HPV6 and HPV11 still being the two most predominant genotypes. As the diversity of HPV has expanded to cover >200 genotypes, the role of other HPV genotypes in RRP has become more relevant. Omland et al. (2014) screened 221 patients (174 AO-RRP and 50 JO-RRP) for HPV genotypes 6, 11, 16, 18, 31, 33, 35, 39, 45, 51, 52, 56, 58, 59, 68a, and 68b [64]. In total, 93.7% (207/221) of the RRP samples were HPV-positive. HPV6 was the most prevalent (133/207 HPV+ samples), followed by HPV11 (40/207) and HPV 6/11 co-infections in 15/207 cases. HPV6 or HPV11 co-infections with one or two HR-HPV types were found in 19/207 patients. HPV33 was the most prevalent genotype (*n* = 14) in these co-infections, followed by HPV45 (*n* = 3), HPV18 (*n* = 2), HPV16 (*n* = 1), HPV31 (*n* = 1), and HPV 35 (*n* = 1). Notably, 14 patients (6.3%) were negative for any tested HPVs, and 6 of them developed SCC. Using metagenomic sequencing, one patient also proved to have HPV8, which is a cutaneous HR-HPV. Co-infection with HR-HPV or being HPV-negative was significantly more common among the patients with AO-RRPs than in those with JO-RRPs. Interestingly, HPV11 was more prevalent in children than in adults [64].

Recently, Hoesli et al. (2020) studied 184 patients fulfilling the following strict inclusion criteria of RRP: (1) Visually obvious papillomas, (2) recurrence requiring multiple surgeries, and (3) histologically confirmed diagnosis of an LP [11]. Altogether, 87.0% of the lesions (160/184) were LR-HPV positive, while 9.2% harbored the following other genotypes: HPV16, 18, 31, 44, 45, 55, and 70. Four patients (2.2%) had co-infections in different combinations: HPV11 with HPV16, HPV76 or HPV84, and an HPV18 with HPV45. RRPs of three patients (3/17) with non-HPV6 or non-HPV11 progressed to laryngeal SCC. In total, 3.8% (7/184) of the RRP lesions were HPV-negative, of which 2/7 progressed to carcinoma. Similarly, Omland et al. (2014) reported that high-grade laryngeal neoplasia was found more frequently in HPV-negative than in HPV-positive lesions (RR = 2.35, 95% CI 1.1–4.99); moreover, HPV-negative RRP biopsies occurred more frequently in adult-onset 526 patients and were associated with an increased risk of laryngeal carcinoma [64].

These data implicate that a minority of RRPs are HPV-negative, which, however, can also progress to malignant lesions. Clearly, RRPs can be caused by HPV genotypes other than HPV6/11, and a minority of those can undergo malignant transformation.

### 6.2. Factors Associated with RRP and Its Outcome

#### 6.2.1. Acquisition of RRP

Kashima et al. (1992) concluded that JO-RRP patients were more often firstborn, delivered vaginally, and born to teenage mothers [65]. On the contrary, AO-RRPs were associated with sexual habits of the patients such as higher number of lifetime sex partners and frequency of oral sex [65]. Niyibizi et al. (2014) conducted a systematic review covering 2296 JO-RRP cases [15]. The identified risk factors confirmed a maternal association, and were linked with birth history and genital warts during pregnancy and delivery. However, some immune response-related factors were also associated with JO-RRP acquisition and disease severity.

Recent evidence supports the view that there is an immunologic tolerance to HPV6 and HPV11 in patients with JO-RRP [66,67]. Additionally, HLA-DRB1*0301 and DQB1*0201 have been associated with reduced interferon-γ expression in patients with RRP [68,69]. When HPV infects the epithelium of susceptible individuals, HPV oncoproteins, predominately E6, can interfere with innate immunity and skew the adaptive immune response to a type 2 T helper cell (Th2)-like or T regulatory (Treg) cell phenotype [66,67,68]. In line with this, we also found in our Finnish Family HPV Study that HPV16 associated cell-mediated immunity (CMI) was shifted towards cytokines IL-5 (type 2 cytokine) and IL-10 and IL-17A in children born with an HPV DNA-positive placenta and/or HPV-positive cord blood [69,70]. We also identified two children without any HPV16-specific CMI response or antibodies, even if they had persistent oral HPV16 infection. Thus, the cytokine profile in children infected with HPV16 during early life suggests that the viral dose and/or specific environment created by the placenta may have a significant impact on the type of HPV-specific immunity.

In conclusion, research is accumulating to support the notion that the RRP microenvironment is immunocompromised; thus, regaining Th1 cell function may be a durable approach to prevent persistent infection.

#### 6.2.2. JO-RRP and HPV11 Are Prognostic Factors of RPP Outcome

The outcomes of RRP are variable among individual patients, and several studies have focused on the risk factors associated with a more aggressive outcome. Most studies agree that a younger age at onset of JO-RRP is an important independent predictor of disease severity [10,15,16,64]. Age at onset was the only significant risk factor for aggressive disease in JO-RRP, whereas in adults, both HPV11 and observation time ≥10 years were significant [64]. In a series by Costa et al. (2019), 36 patients with JO-RRP and 44 patients AO-RRP were subdivided into low- and high-risk groups based on Derkay scores [71]. HPV6 detection was significantly higher in AO-RRP cases (*p* = 0.03), whereas HPV11 was more prevalent in JO-RRP cases (*p* = 0.02), and even more prevalent in high-risk JO-RRP cases (Derkay laryngoscopic scale ≥ 20) (*p* = 0.04) [71]. Similar results were published by Nogueira et al. (2021), who found that age of onset of RRP had an important impact on the number of surgeries needed to control the disease. Patients with JO-RRP and HPV11 tend to present an increasingly worse Derkay score at each surgery. HPV genotype among the AO-RPP patients had no impact on disease outcome [10].

#### 6.2.3. Physical State of HPV6 and HPV11, Their Variants and Other Viral Factors in RRP

Studies on the genetic variants or viral integration of HPV6 and HPV11 genotypes are lacking, and the results from scattered studies have been disappointing. Do Bonfim et al. (2015) reported that JO-RRP lesions harbored exclusively HPV-6vc-related variants of the five variants studied, whereas, among AO-RRP, HPV-6a variants were more prevalent. The HPV-6vc reference was more transcriptionally active than the HPV-6a reference [72]. Ikegami et al. (2021) conducted a phylogenetic analysis of 23 HPV6-infected LPs obtained from 13 patients [73]. The authors identified three different HPV6 subtypes: HPV6a, HPV6b, and HPV6vc, with the latter being the most common. HPV6 subtype or viral load/HPV mRNA expression levels were not associated with the clinical parameters (age, sex, tumor location, Derkay score) [73], as also recently confirmed by two other studies [10,74]. Sichero L et al. (2021) found no association between any particular HPV11 variant and clinic-pathological features among the 79 RRP patients studied [74]. Among the HPV11-positive patients, only variants from the A1 and A2 sub-lineages were detected, with A2 (62.5%) variants being most prevalent [74].

In 2012, Ure and Forslund analyzed the methylation status of the HPV6 upstream regulatory region (URR) in SCPs from the upper aero-digestive tract of six adult patients. All CpGs in the URR were non-methylated, from both basal/intermediate and superficial cells, suggesting that methylation is not involved in the regulation of transcription from the HPV6 URR, regardless of epithelial differentiation [75]. Similarly, Deng et al. (2017) studied the methylation status of HPV6 URR in 15 HPV-positive AO-RRP specimens derived from 9 patients. The viral copy numbers of HPV6 were high and the physical state of the virus was always episomal [76]. Hypo-methylation and scattered patterns of methylated CpGs at the URR of HPV6 were identified without any association with disease outcome [76]. Based on the current evidence, integration of HPV6 and HPV11 is an extremely rare event in RRP, but it can be found anecdotally in RRP-derived carcinomas [77,78,79].

#### 6.2.4. Risk Factors for Distal Spread

Distal spread of RPP occurs in 5% to 48% of cases. Various hypotheses have been proposed to explain this distal spread, including the extension of papillomas by contiguity, diffuse viral contamination, and iatrogenic factors, such as laryngoscopy, bronchoscopy, tracheostomy, and surgical manipulation. Risk factors for the spread of RRP toward the lower respiratory tract include HPV11 infection, age younger than 5 years, tracheostomy performed to avoid airway obstruction, and previous invasive procedures [80,81]. Gélinas et al. (2008) reviewed cohort studies on RRP, and 3.3% (55/1666) of the patients) had pulmonary involvement [82]. In the same study, they also evaluated cross-sectional studies, which showed that 2.3% of the patients (28/1202) had a pulmonary dissemination. Pulmonary involvement in RRP was twice as frequent among men than women, and was associated with a more aggressive clinical course. The median interval between the diagnosis of RRP and lung involvement was 8 years (range < 1–45) [82].

#### 6.2.5. Malignant Transformation

Malignant transformation (to SCC) in AO-RRP occurs in 3% to 6% of the patients, while it is rarer in JO-RRP patients, <1%. As expected with JO-RPP, malignant transformation occurs in older children; the median time between the diagnosis of JO-RRP and the diagnosis of SCC is 19 years (range 4–45 years) [83]. HPV11 seems to have a higher potential for malignant transformation than HPV6 [64,65,78,84]. Age of disease onset is the strongest predictor of dysplastic transformation in the adult and pediatric populations. Carcinoma in a pre-existing RPP was uniformly associated with pulmonary disease in the JO-RRP population [82,83].

Karatayli-Ozgursoy et al. (2016) performed a study on 159 RRP patients, comprising 96 AO-RRP and 63 JO-RRP patients [85]. In their cohort, 139 (87%) had benign LP as the only histological diagnosis. Among the AO-RRP cohort, 10 patients (10%) were diagnosed with dysplasia or carcinoma in situ (CIS) in addition to LP, and five patients (5%) had undergone malignant transformation to SCC. The patients with dysplasia or SCC had a higher age of disease onset compared to those devoid of dysplasia or SCC (56 vs. 45 years old; *p* = 0.0005). Of the 63 JO-RRP patients, there were no cases of dysplasia, but 3 (5%) patients developed an invasive SCC, all with pulmonary involvement. The JO-RRP patients with SCC had a younger age of disease onset (2 vs. 6 years; *p* = 0.009) and a higher rate of tracheal involvement than those who did not develop SCC. Gender, smoking history, number of operations, or use of cidofovir showed no association with the development of dysplasia or SCC in either the AO-RRP or the JO-RRP cohort [85].

Until recently, malignant transformation had been reported only for HPV11-associated RRP in 2–4% of all RRP-cases, but not for HPV6. However, Huebbers et al. described a novel molecular mechanism in the first case of HPV6-associated laryngeal SCC identified in JO-RRP. HPV6 was shown to be integrated in the aldo-keto reductase 1C3 gene (AKR1C3) on chromosome *10p15.1*, resulting in the loss of this gene’s expression. Alterations in AKR1C gene expression have previously been implicated in the pathogenesis of other HPV-related malignancies [86].

### 6.3. RRP Can Be Prevented by HPV Vaccinations Targeting HPV6 and 11 (Gardasil)

Reliable data on the efficacy of HPV vaccines in preventing SCP in the aero-digestive tract are available only for RRP. The off-label HPV vaccines used for treatment of these lesions fall outside the scope of this discourse. Australia is the pioneer and model country in implementation of national HPV vaccination programs. This is because the 4-valent vaccine introduced in 2007 for girls and 2013 for boys has had an excellent uptake, with over 80% among girls and 75% among boys, with at least two doses. Since October 2011, surveillance of JO-RRP has been conducted by the the Australian Pediatric Surveillance Unit (APSU) [87]. Thus far, no other pediatric SCPs in the head and neck region have been monitored. The first 5-year survey covered the period 2012–2016. The average annual incidence rate of JO-RRP was 0.07 per 100.000. The largest number of cases was reported in the first year, with a decreasing annual incidence thereafter. Rates declined from 0.16 per 100.000 in 2012 to 0.02 per 100.000 in 2016 (*p* = 0.034). Among the 15 incident cases of JO-RPP (60% males), none of the mothers were vaccinated before pregnancy, 20% had maternal history of genital warts, 60% were first-born, and 13/15 were born vaginally. Of the examined cases, only HPV6 (*n* = 4) or HPV11 (*n* = 3) were detected. Of the two cases reported in 2017, one was a probable case of non-laryngeal JO-RRP and the other occurred in a child who was not born in Australia. In 2019, no cases of JO-RRP were reported to the APSU in any Australian state or territory for a second consecutive year. Thus, in the eight years of JO-RRP surveillance, seventeen confirmed cases were reported, with one to four cases identified per year until the end of 2017 [88].

Taken together, the widespread routine HPV vaccination (4-valent Gardasil) of both girls and boys has interrupted community circulation of the causal HPV types of JO-RRP and has consequently dramatically reduced the risk of perinatal exposure and subsequent disease in Australia. It remains to be seen whether the effect is equally dramatic for the other HPV-associated pediatric SCPs.

## 7. Tracheal Papillomas (TP)

Tracheal papilloma (TP) is a distinct rarity, mostly manifested as the tracheal involvement of RRP. TPs have both a juvenile and an adult onset, presenting with either a JO- or AO-papillomatosis, mostly due to the distal spread of RRP from the larynx [80,81,82]. TPs are more common (3–26%) than pulmonary involvement (1–3%). Tracheotomy is also associated with progression of LP into the trachea. However, RRP arising in the trachea without a laryngeal lesion has been occasionally reported both in adults and children [89,90].

The presence of TPs has also been associated with primary biliary cirrhosis, Cowden disease, and tuberculosis [91,92,93]. TPs may also resolve spontaneously. However, recurrences occur and may be massive and rapid, leading to airway obstruction, tracheostomy, or laryngectomy. Recurrences often terminate at puberty. Similar to RRP, most cases of TP are caused by HPV6 and/or HPV11 infections but, occasionally, HPV16 and HPV18 have been identified [94,95]. Smoking, age above 40 years, and infection with HPV16 and HPV18 are risk factors for malignant transformation. TP can occur at any age and in all ethnic groups. The AO-TP affects males and females in a 4:1 ratio.

Interestingly, TPs associated with laryngeal lesions have low incidence of malignant transformation, in contrast to patients whose lesions are limited to trachea and bronchi, usually manifested in adulthood, and showing a higher incidence of malignant transformation [93,94,95]. Abramson et al. [95] showed that HPV infects tracheal mucosa and is maintained as a latent infection in the trachea as efficiently as it is in the larynx. Therefore, they suggested that the low frequency of tracheal disease reflects a lower frequency of HPV activation. They also proposed that cellular factors that differ between the stratified squamous epithelium of the larynx and the ciliated pseudo-stratified columnar epithelium of the trachea contribute to this difference [95].

## 8. Esophageal Squamous Cell Papilloma (ESP)

Esophageal squamous cell papillomas (ESP) are rare tumors that were first described in 1928 by Patterson et al. [96] and histologically verified as a distinct entity in 1959 by Adler et al. [97]. Since then, the literature on ESPs increased slowly by reports on single cases and a small case series until the early 1980s, followed by a significant burst thereafter. Until 1994, only 141 cases had been reported; by 1998, this number had increased to 218 [56]. During the 2000s, the literature on ESPs increased with speed, and it currently covers several hundreds of cases reported as large clinical series.

ESP is detected mostly incidentally by esophagogastroduodenoscopies in 0.01–0.45% of cases. However, although ESP is rare, its prevalence has been increasing; Pantham et al. (2017) reported an increase in ESP prevalence from 0.13% in 2000 to 0.57% in 2013 [98]. ESP is most commonly diagnosed in patients aged between 43 and 50 years, and the male–female ratio is variable. ESPs are usually solitary (85%) and located in the distal esophagus (70%) but have been reported as multiple lesions or, in a few cases, papillomatosis. They are usually small in size, ranging between 2 and 6 mm (Figure 4). However, a number of well-documented multiple ESPs have been reported mostly in children and not infrequently, and are associated with RPP [99]. The clinical course of ESP is benign; only anecdotal cases of malignant transformation are reported in the literature, suggesting that such an event must be extremely rare. On the other hand, there is some emerging evidence that esophageal papillomatosis can regress spontaneously.

### 8.1. Etiology of ESP

The exact etiology of ESP is still uncertain, but several etiologic factors have been proposed such as chemical factors, mechanical factors, and infections with HPV and/or EBV [99,100]. The suspected chemical and mechanical factors result in mucosal injury with a hyper-regenerative response such as in gastro-esophageal reflux disease (GERD). This may explain why two-thirds of the reported cases of ESP have been localized to the lower third of the esophagus, a site exposed to chronic irritation from gastric acid reflux. Other reported sources of trauma include mechanical sources (e.g., nasogastric tubes and previous gastroesophageal surgeries). It is also associated with syndromes such as Goltz–Gorlin syndrome, angioma serpiginosum, and Cowden syndrome [101,102].

#### HPV and ESP

The role of HPV infection in ESP is still a controversial issue despite the fact that 40 years have elapsed since our original observations in the early 1980s. We performed a systematic review and meta-analysis of the literature reporting on HPV detection ESP, covering the literature through to May 2012. In total, 39 studies were eligible, covering 427 ESPs from different geographic regions [103]. Altogether, the HPV detection rate in ESP was 30.9%, (132/427), but variation was wide. In stratified meta-analyses and meta-regressions, variability in HPV detection rates in ESPs was not explained by either HPV detection methods or geographic origin in the study [103]. Thus, we could not confirm these frequently presented explanations for the wide variation in HPV detection in ESP [99].

Tiftikçi et al. (2017) analyzed HPV in ESPs diagnosed in 21 women and 17 men with a mean age of 41 years (range 17–67 years). Most of the ESPs were located at mid-esophagus (68%) [104]. Eight of the thirty-eight patients (21%) had associated erosive esophagitis, and fourteen patients (36.8%) had *Helicobacter Pylori* (H. pylori). Altogether, 19% (7/38) of the ESP analyzed were positive for HPV DNA; three for HPV6 and one each for HPV16, 18, 31, and 81. There was no correlation between the presence of HPV and patient’s age, reflux esophagitis, H. pylori, smoking habits, or location of ESP [104].

An association between HPV and esophageal adenocarcinoma (EAC) and Barret’s esophagus (BE) has been reported, as was summarized in a recent systemic review by Kunzmann et al. 2017, which examined a total of 30 studies [105]. The pooled prevalence of HPV in EAC samples was 13% (*n* = 19 studies, 95% CI: 2–29%) and 26% (*n* = 6 studies, 95% CI: 3–59%) in BE samples. HPV prevalence was higher in EAC tissue than in esophageal tissue from healthy controls (*n* = 5 studies, pooled OR = 3.31, 95% CI: 1.15–9.50). The prevalence of EBV in EAC was 6% (*n* = 5, 95% CI: 0–27%) [105].

ESPs have also been detected in children, as found for all SCPs in different sites of the aero-digestive tract. Recently, Tou and Al-Nimr (2021) completed a retrospective search through all endoscopies (EGD) performed (with various indications) in children under 18 years of age from 2000 to 2014 in their pediatric hospital [106]. Of the 12,459 children who required an EGD, 10 children were identified with ESP in the biopsy, with ages ranging from 2 to 17 years. This provides an estimated prevalence of 0.08% over the entire study period. In total 60% of the detected lesions were in the proximal esophagus, and 80% of the patients had isolated lesions. However, none of these lesions tested HPV-positive with fluorescence ISH.

## 9. HPV Can Be Acquired at Early Age

It is currently well established that HPV infection is not exclusively a sexually transmitted infection (STI). HPV infection can be acquired at an early age or even during birth, as suggested by the presence of HPV in the placenta, amniotic fluid, and cord blood (for review see [107,108,109]). Vertical transmission can be categorized as peri-conceptual [110], prenatal (during pregnancy) [69,111,112], and perinatal (during birth or immediately thereafter) [112,113,114,115].

A feature common to SCPs in the aero-digestive tract is that they all present a bimodal incidence and prevalence pattern, being present in children and adults irrespective of their anatomic site. Thus far, however, only JO-RRP has been closely associated with maternal HPV transmission, though HPV can also infect both oral and pharyngeal mucosa vertically. This raises important questions challenging the future HPV studies: (1) can HPV spread from oral and pharyngeal sites distally to the larynx and esophagus? (2) Can HPV acquired at an early age become a latent HPV infection which becomes activated later in life in these sites? (3) Is HPV transmission via blood possible, as recently suggested by animal studies? These important issues are discussed here, and because animal studies have also shed light on this topic, they will be addressed in Section 10.

### 9.1. Mother Is the Main Transmitter of Her Offspring

A meta-analysis of 3128 mother–child pairs confirmed that children born to HPV-positive mothers are 33% more likely to be HPV-positive than children born to HPV-negative mothers are. This risk was even higher (45%) when only HR-HPV infections were considered [116]. A recent systematic review on intra-uterine HPV transmission resulted in a pooled percentage of 4.9% (95% CI 1.65–9.85%) for antenatal vertical HPV transmission, and, importantly, the mode of delivery had no effect on this transmission [117]. Another meta-analysis by Chatzistamatiou et al. (2016) [118] provided contradictory evidence, while caesarean section decreased the risk for perinatal HPV transmission by approximately 46%. Notably, however, perinatal transmission still occurred in approximately 15% of the children born by caesarean section. In our recent study on oral HPV-infection in children followed during their six first years of life, we found that both mother’s baseline oral HPV carriage (OR 1.92, 95% CI 1.35–2.74) and HR-HPV seroconversion (OR 1.60, 95% CI 1.02–2.50) were associated with persistent oral HR-HPV infection in children [119]. Furthermore, HLA-G molecules might have a role in predicting a newborn’s likelihood of developing oral HPV infection at birth [120].

### 9.2. Perinatal Transmission of Cutaneous HPV into Nasopharynx Is Additionally Possible

The transmission of cutaneous HPV genotypes from a mother to her newborn is thought to occur through skin-to-skin contact and during breastfeeding. Recently, Dassi et al. (2020) collected nasopharyngeal specimens from 0–12-month-old infants born by vaginal delivery (and breastfed at the time of sample collection) to investigate the perinatal transmission of α- and β-HPVs [121]. In total, 14 out of 113 (12.4%) samples tested positive for HPV and sequence analysis identified 8 β-genotypes (HPV 5b, 20, 25, 100, 107, 124, 152, and RTRX7). The authors also performed a comprehensive review of the published studies on the prevalence of mucosal and cutaneous HPVs. Among 5126 newborns covered by these studies, 10% and 53% were positive for α- and β-HPVs, respectively. In all studies, there was an inverse correlation between the rate of α- HPV positivity and age, while a significant positive trend was observed in β-HPV detection and age, with the highest rate among children > 12 months of age (*p* < 0.001). Currently, the natural history of β-HPVs in children is practically unknown. However, though not widely studied yet, HPV-genotypes belonging to either β-, or γ-genera have been detected in sinonasal-, oral- and pharyngeal SCPs. [121]. More studies are clearly warranted to explore the role of β- and γ-HPV infections in the upper-aero-digestive tract, particularly in early childhood.

### 9.3. Outcome of Vertically Transmitted HPV Only Partly Known

The next pertinent question is, what is the outcome of vertically transmitted α-HPV infections? We recently published data on oral HPV infections during the first 6 years of children born to 329 mothers included in the Prospective Finnish Family HPV Study (ongoing since 1989: *n* = 331 children; *n* = 329 mothers; and *n* = 131 fathers). The prevalence of oral HPV in these children varied from 8.7% to 22.8% over time, being highest at birth and lowest at the 3-year follow-up visit. Altogether, 18 different HPV genotypes were identified in the oral mucosa, of which HPV16 was the most prevalent, followed by HPV18, 6, 33, and 31. The prevalence of multiple-type infections varied from 0.3% to 3.7%. Only 41.4% (135/329) of the children remained HPV-negative for all oral samples (nine samples altogether), collected during the entire 6-year follow-up period [119].

None of the HPV genotypes present at birth seemed to promote the acquisition of another specific HPV genotype, not even an HPV from the same clade. However, newborns with oral HPV6 or HPV11 (*n* = 4) acquired only HPV16 or HPV18 genotypes during the follow-up. When analyzed by gender, HPV prevalence was higher at 1-, 2-, 12-, and 36-month visits in boys, but not at the 6-year visit. Importantly, HPV positivity at birth or later was unrelated to the mode of delivery. The results showed that, during the 6-year follow-up, 63% (26/41) of these children carried the same genotype that was already detectable at birth, including 4/11 children positive with HPV6 at birth who still had this genotype at their 6-year follow-up visit [119]. Interestingly, our unpublished data showed that firstborn children had lower levels of maternal antibodies to HPV6 and HPV11 than non-first born children. Such differences among the children were not found for HPV16-, HPV18, or HPV45 antibodies.

In another prospective study setting, Puranen et al. (1997) [122,123] examined children 0.3–11.6 years of age and who were born to mothers prospectively followed-up for genital HPV infections with comprehensive HPV data available at the time of delivery. In this cohort, 31.6% (31/98) of the oral brush samples were HPV-positive, of which 11 children had the same genotype as found in the mother’s genital samples at delivery. Oral HPV DNA was more prevalent among children aged less than 6 years. Clinical lesions in the oral mucosa were found in 22.4% (22/98) of the children, all being hyperplastic and mostly inconspicuous. However, only 8 of these 22 children with tiny oral lesions (36.4%) had an HPV-positive oral brush sample. One palatinal papillary lesion excised from a 7-year-old girl was a typical OP on light microscopy, and was confirmed to be HPV16-positive by ISH. Her mother’s genital sample was also HPV16-positive at the time of delivery [122,123].

## 10. Animal Models to Study SCPs of the Head and Neck

HPVs cannot be directly studied in animal models because PVs are strictly host species-specific. However, several animal models are useful for studying PV biology and immunity, as reviewed by Doorbar in 2016 [79]. notably, two rabbit papillomavirus models have been widely used to understand the behavior of HPV. Cottontail rabbit papillomavirus (CRPV) is a cutaneous-tropic virus whose lesions spontaneously progress to cancer, while rabbit oral papillomavirus (ROPV) is a mucosa-tropic virus that induces oral infections. ROPV had already been recognized in 1943 by Parsons and Kidd [14], who also described the natural history of OPs caused by this virus. Later, it was shown that ROPV had a tissue tropism and a life cycle organization that resembled those of the human mucosal types (HPV11 and HPV16). Thus, ROPV appears most appropriate for studies of the life cycles of mucosal PVs [14,79,84,124].

### 10.1. Rabbit Oral Papillomavirus (ROPV)—A Model to Study the Outcome of Mucosal Papillomas and Their Infectivity

Natural history studies on HPV infections of SCPs in the head and neck are lacking. However, the outcome of oral HPV infections most probably mimics that described in the early experimental study by Parsons and Kidd in 1943 [14]. Thus, it is worthwhile to review that milestone study in this context. Close contacts are known to be important in viral transmission, as shown also in the Parsons and Kidd study with domestic rabbits (university animal house and wild rabbits). Natural growths were present in 16.3% of all domestic rabbits (119/722) examined, and in 9.6% of the 311 normal young adult rabbits, many of which were bought outside the institute. Additionally, 3.6% of 273 institute-bred rabbits less than 4 months old had OPs, which were histologically typical SCPs. These OPs in the domestic rabbits were located under the surface of the tongue, and occasionally in the gums or on the floor of the mouth. However, none of the 300 wild rabbits had OPs, as their contact with other rabbits in nature was only infrequent.

Experimental transmission of OPs was successful in 81.5% (66/81) of the rabbits using crude suspensions of Bekerfeld filtrates of ground papillomas. New papillomas developed after 6–36 (average 14) days, increased in size for about 1 month, and some papillomas persisted for as long as 400 days. The isolated virus, ROPV, was stable and a high temperature (75 °C to 80 °C) for 30 min was needed to inhibit viral infectivity. Rabbits with OPs already regressed or undergoing regression were resistant to ROPV re-infection. In some of the rabbits, the virus remained latent after clearance of the papilloma but could be activated simply by injuring or irritating the area. Viruses could be washed out of the mouths in the rabbits carrying the growths and from the mouths of rabbits with normal mucosa but carrying a latent infection. Transmission of the virus from a mother to her offspring was shown, and there were “papilloma families”, in which transmission of the virus from a mother to her offspring occurred via saliva during licking [14].

A similar situation is seen in the multimammate mouse (Mastomys natalensis/Mastomys coucha), in which latent MnPV (Mastomys natalensis papillomavirus) acquired early in the animal’s life can subsequently be activated to produce papillomas, keratoacanthomas, and other skin tumors [125].

### 10.2. Latency and Papillomavirus Infections

Activation of latent HPV infection could be the most likely source of primary and recurrent papilloma in the aero-digestive tract, as it is in the genital tract. Several animal studies have confirmed the ability of CRPV to persist at low DNA levels in clinically normal epithelium without visible disease. However, mild mechanical irritation or exposure to ultraviolet light could activate the virus, leading to the emergence of clinical lesions [124,125,126], as also shown with the ROPV model [14]. Importantly, asymptomatic infection was shown to be associated with the production of E1 transcripts, which are needed for the stable maintenance of viral genomes in infected epithelial cells [127]. Most CRPV-induced papillomas do not spontaneously regress, but when complete regression does occur, CRPV DNA can be detected in clinically normal tissues. A strong CD8+ response to virus-infected cells is needed to suppress the latent infection [128]. Similarly, also ROPV was shown to persist in the absence of clinical and microscopic disease for up to a year following the resolution of OP [46]. The results suggested that ROPV genomes were maintained as a latent infection, and the site of latency was a subset of basal epithelial cells at sites of previous experimental infection [46]. Persistence of viral DNA in the epithelial basal layer suggests a model for PV latency following immune regression [46].

### 10.3. Transmission of Papillomaviruses via Blood

The evidence for HPV in blood mononuclear cells of women with genital HPV lesions was presented in 1991 [129], but these results were totally ignored at the time. More recently, Cladel et al. (2019) provided direct evidence on blood transmission of PVs in rabbit and mouse models [130]. Blood infected with PV yielded infections at permissive sites with detectable viral DNA, RNA transcripts, and protein products. Due to the importance of this study, the seven main conclusions are re-iterated here, as given by the authors: (1) Viral sequences could be detected in the blood of infected animals; (2) virus introduced into the blood yielded tumors at both cutaneous and mucosal sites; (3) CRPV DNA introduced into the blood yielded papillomas at prepared skin sites; (4) similar mechanisms are used for infections via the blood and by direct application of virus to the skin, as determined by RNAseq analysis; (5) transfusion of blood from an animal that had received virus via intravenous infection to a naïve sibling resulted in papillomas in the transfusion recipient; (6) virus introduced via intravenous delivery yielded infections in the stomach, as well as in normally permissive sites; (7) blood from animals with active infections could induce infections in naïve mice when transfused into these animal [130].

HPV transmission by blood represents a conceptually novel idea and, if accepted more generally, this would change the current thinking about the modes of HPV spread within the host.

## 11. Conclusions

SCP in the upper aero-digestive tract is a rare disease entity, as summarized in Table 1. The bimodal age presentation both at childhood and in adults is characteristic for all these papillomas, although SNPs and ESPs are extremely rare among children. A male predominance is found for SNPs (both SNIP and SNEP) and AO-RRP. The most widely studied entities are SNPs and RRP of the larynx. This is because of their more aggressive behavior and potential for malignant transformation. Aggressive clinical course of the disease is found in 20% of the JO-RRP cases, while malignant transformation is most prevalent among patients with SNIPs (7–11%), followed by RRP (3–6%).

HPV6 and HPV11 are the two major HPV genotypes present in all papillomas, irrespective of their anatomic location. However, HPV detection rate is highest in RRP, followed by exophytic SNP, OP, and SNIP. Importantly, although HPV association is highest with RRP, there is always a minor proportion of RRPs that tests HPV-negative (4–6%), and these HPV-negative papillomas can also progress to malignancy. Recent evidence has also shown that not only mucosal α-HPV genotypes but also cutaneous HPVs from the β- and γ-genera are detectable in sinonasal-, nasopharyngeal-, oropharyngeal-, and oral papillomas; however, until now, only one RRP case has been reported as testing HPV8-positive.

HPV diversity in these lesions is much wider than previously considered, as summarized in Table 1. As an example, 13 different HPV genotypes have been identified in RRPs. Malignant transformation of SNIPs is associated with HR-HPVs, as signified by the LR-HR ratio in lesions at different staged of progression: 4.8:1 for SNIP, 1:1 for SNIP with dysplasia, and 1:2.8 for SNIP with carcinoma. Malignant progression of RRP is associated with HPV11 infection, together with age at the onset of JO-RRP or AO-RRP. It is currently unknown why LR-HPV6 and HPV11 play such key roles in RRP pathogenesis and disease outcomes. One hypothesis is that RRP could be a multigene disease, in which the HPV genotype and tissue-specific immune deficiency prevent an effective clearance and control of these LR-HPV infections [84]. Another aspect to discuss is that, among all aero-digestive tract papillomas, JO-RRP is the only lesion that is intimately connected with maternal and birth history, although at least nasopharynx and the oral mucosa are all exposed to α- and cutaneous HPV genotypes at birth, and even prenatally. It remains to be seen whether maternal HPV immunity, including maternal HPV antibodies, and presence of HPV in the placenta or amniotic fluid, makes this anatomic site more vulnerable to persistent or latent HPV-infections than other sites.

## Figures and Tables

**Figure 1 viruses-13-01624-f001:**
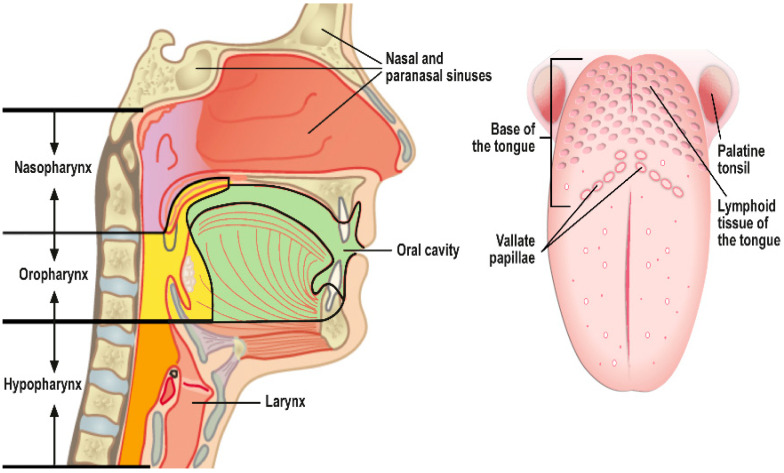
A schematic presentation of the upper aero-digestive tract. This review also includes the more distally located trachea and esophagus. The figure is modified from the original figure published as Figure 3.1, page 16, Report 2/2019, Should boys’ HPV vaccinations be included in the national vaccination programme? the National Institute for Health and Welfare. https://www.julkari.fi/handle/10024/137477, accessed on 23 January 2019. Permission accessed 10 May 2021.

**Figure 2 viruses-13-01624-f002:**
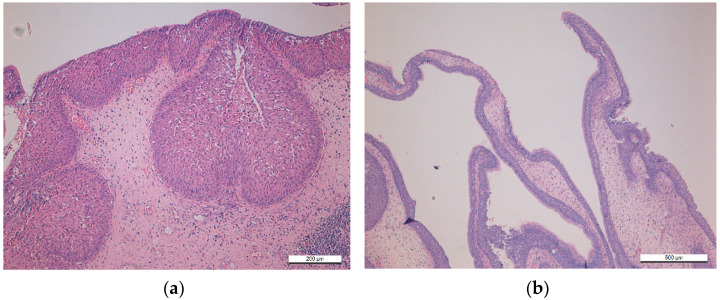
Two different histological types of sinonasal squamous cell papillomas (SNP): (**a**) sinonasal inverted papilloma (SNIP); (**b**) sinonasal exophytic papilloma (SNEP).

**Figure 3 viruses-13-01624-f003:**
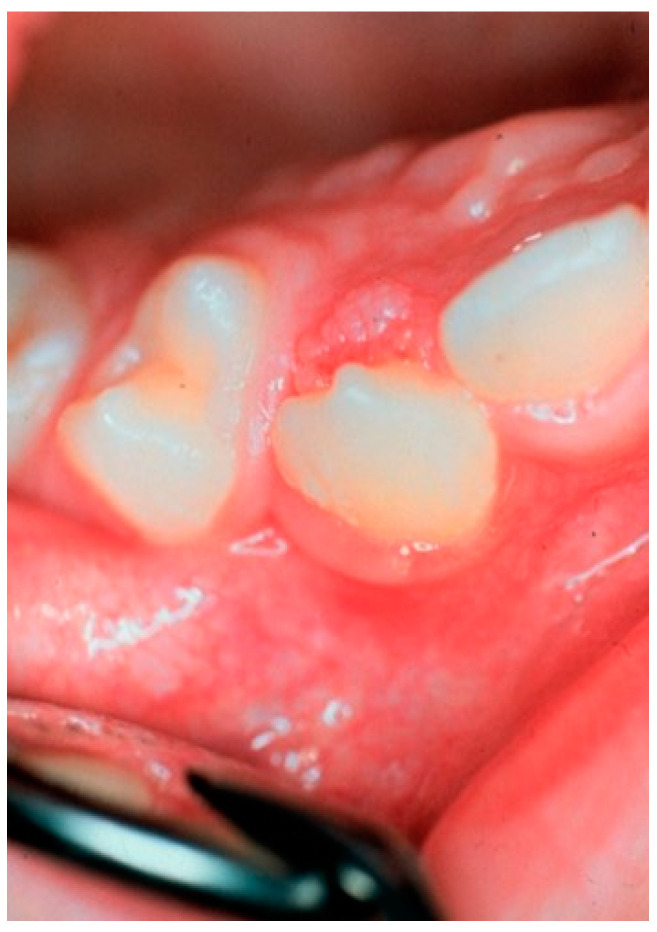
HPV11-positive oral papilloma of a child.

**Figure 4 viruses-13-01624-f004:**
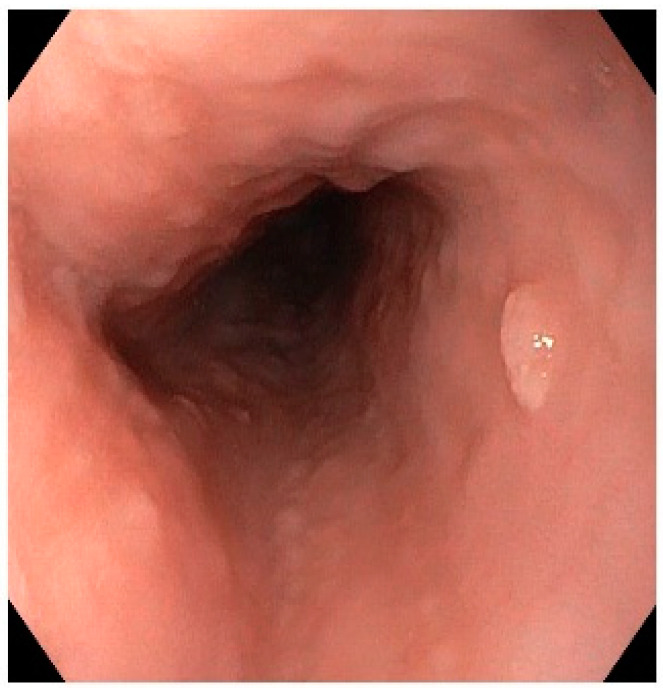
An endoscopic view of a typical esophageal papilloma (by courtesy of Prof. Matti Eskelinen, University of Eastern Finland, Kuopio).

**Table 1 viruses-13-01624-t001:** Summary of the squamous cell papillomas in the upper aero-digestive tract.

Characteristic	Sinonasal Papillomas: Inverted/Exophytic/Oncocytic	Oral Papillomas	Laryngeal Papillomas	Esophageal Papillomas
Frequency Proportion among sinonasal suptypes	Sinonasal overall 0.74–2.3/100,000 children extremely low incidence SNIP 65–70% SNEP 20–25% SNOP 3–6%	children common, but prevalence unknown adults 0.5/100,000	children 0.6–4.3/100,000 adults 1.8–2.3/100,000	children 0.08% adults 0.01–0.45%
Common location	SNIP nasal cavity and maxillary sinus SNEP lower anterior nasal septum SNOP lateral nasal wall	tongue, gingiva, under lip (oropharynx: soft palate, uvula, tonsil)	vocal cords and ventricles, false cords	proximal esophagus
Male to female ratio	children 3:1 SNIP 2.5–3:1 SNEP 2–10:1 SNOP 1:15	1:1	children 1:1 adults 3:2	children 1:2.3 adults 5:2 to 1:5
Age at presentation	children 6–15 years adults: SNIP 50–60 years SNEP 30–50 years SNOP > 50 years	children adults 30–50 years	JO-RRP < 5–8 years AO-RRP 20–40 years	children 2–17 years adults 40–50 years
Prevalence of HPV	SNIP 25–39% SNEP 65.3% SNOP 22.5%	overall 27–48% HPV+ samples: 48% alpha-HPV 23% gamma-HPV	76%-93.7%	10.5–57%
Mucosal Alpha HPVs low-risk high risk	HPV6, 11, 42 HPV16, 58, 83	HPV6, HPV11 HPV16, 18, 35, 51, 74	HPV6, 11 HPV16,18, 31, 33, 35, HPV45, 55, 70, 76, 84	HPV6 HPV16,18, 31, 81
Cutaneous HPV beta-HPV gamma-HPV	NA	HPV5, HPV12, HPV22, HPV23, HPV120 HPV121, HPV123, HPV130, HPV131, HPV161	HPV8 NA	NA
Recurrence	SNIP 25–30% SNEP 22–50% SNOP 39%	unusual approx. 4.1%	aggressive course in 20% of JO-RRP, distal spread 5–48%	NA
Risk of malignant transformation	incidence 0.38/10^6^ SNIP 7–11% SNEP SNOP	no evidence	JO-RRP < 1% AO-RRP 3–6%	rare

## Data Availability

Not applicable.

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
