# Peer review of "HPV-Associated Benign Squamous Cell Papillomas in the Upper Aero-Digestive Tract and Their Malignant Potential"

_viruses, 2021, doi:10.3390/v13081624_

Round 1

Reviewer 1 Report

This review by Stina Syrjänen , and Kari Syrjänen  is interesting by filling an information gap in the pathology of the upper aero-digestive tract represented by papillomatous lesions. The review is well done and I have only minor concerns.

In general, I prefer an indirect way to present data avoiding use of sentences like “We suggested HPV as the potential etiological agent of SNPs in 1983 and 1987”; however this is only a my personal point of view.

On the contrary, I believe that heading "11. RRP can be prevented by HPV vaccinations targeting HPV6 and 11 (Gardasil)" should be moved to heading "6. Laryngeal squamous cell papillomas (RRP)" as a sub-heading. In the present version results regarding efficacy of Gardasil on RRP appear disconnected and isolated from the context while they are more consequential if introduced in the part dedicated to the RRP.

Reviewer 2 Report

This is an interesting review that describes HPV-associated squamous cell papilloma in the upper-aero-digestive tract. Before accept, there are several points to be clarified.

  1. Lines 200-211, page 5. 2.3. Malignant transformation. Several important reports are related to malignant transformation in IP, especially EGFR genetic mutation.  Authors should discuss the findings in addition to HPV infection.
  2. Line 102, 223 'ISNP' should be SNIP.
  3. Line 277 'NP'? 
  4. Line 339 (REF) Please cite the reference.
  5. Before line 642, the authors used EP for exophytic papilloma. However, after that, they used ESP instead of EP. 
  6. Line 677. What is 'SQP'? Is it SP?

Reviewer 3 Report

This is a generally well written (many small grammatical errors which would need to be fixed prior to publication) review article on HPV in benign papillomas of the upper aerodigestive tract. The article is very comprehensive and covers a very broad range of topics, anatomic sites and pathologies. The topics covered include: sinonasal, nasopharyngeal, esophageal and oral papillomas, recurrent respiratory papillomatosis of the larynx and trachea, maternal transmission of HPV, animal models of HPV infection, and HPV vaccination. The review is 21 pages long. While these topics are all related to HPV, each one could be a review article itself. Many of the citations are decades old and thus, the article reads more like a book chapter, citing the history of the topic and giving basic information, as opposed to a contemporary review presenting the newest and most cutting-edge data. I would recommend narrowing the focus of the topics and also concentrating on only the newest data which may be of interest to readers who are already familiar with the background, but would like a review of recent advances in the field.  

Round 2

Reviewer 2 Report

The authors made sufficient responses.

Author Response

Thank you for your review and positive feedback.

Reviewer 3 Report

.

Author Response

Thank you for your review. The corrections have now been made.